# Robotic and laparoscopic gynaecological surgery: a prospective multicentre observational cohort study and economic evaluation in England

Andrew McCarthy [1] Dilupa Samarakoon,[1] Joanne Gray,[1] Peter Mcmeekin,[1] Stephen McCarthy,[1] Claire Newton,[2,3] Marielle Nobbenhuis,[4] Jonathan Lippiatt,[5] Jeremy Twigg[6]

For numbered affiliations see end of article.

**Correspondence to**
Andrew McCarthy;
andrew2.mccarthy@
northumbria.ac.uk

## ABSTRACT

**Objective** To compare the health-related quality of life and cost-effectiveness of robot-assisted laparoscopic surgery (RALS) versus conventional 'straight stick' laparoscopic surgery (CLS) in women undergoing hysterectomy as part of their treatment for either suspected or proven gynaecological malignancy.

**Design** Multicentre prospective observational cohort study.

**Setting** Patients aged 16+ undergoing hysterectomy as part of their treatment for gynaecological malignancy at 12 National Health Service (NHS) cancer units and centres in England between August 2017 and February 2020.

**Participants** 275 patients recruited with 159 RALS, 73 CLS eligible for analysis.

**Outcome measures** Primary outcome was the European Organisation for Research and Treatment of Cancer Quality of Life measure (EORTC). Secondary outcomes included EuroQol-5 Dimension (EQ-5D-5L) utility, 6-minute walk test (6MWT), NHS costs using pounds sterling (£) 2018–2019 prices and cost-effectiveness. The cost-effectiveness evaluation compared EQ-5D-5L quality adjusted life years and costs between RALS and CLS.

**Results** No difference identified between RALS and CLS for EORTC, EQ-5D-5L utility and 6MWT. RALS had unadjusted mean cost difference of £556 (95% CI −£314 to £1315) versus CLS and mean quality adjusted life year (QALY) difference of 0.0024 (95% CI −0.00051 to 0.0057), non-parametric incremental cost-effectiveness ratio of £231 667 per QALY. For the adjusted cost-effectiveness analysis, RALS dominated CLS with a mean cost difference of −£188 (95% CI −£1321 to £827) and QALY difference of 0.0024 (95% CI −0.0008 to 0.0057).

**Conclusions** Findings suggest that RALS versus CLS in women undergoing hysterectomy (after adjusting for differences in morbidity) is cost-effective with lower costs and QALYs. Results are highly sensitive to the usage of robotic hardware with higher usage increasing the probability of cost-effectiveness. Non-inferiority randomised controlled trial would be of benefit to decision-makers to provide further evidence on the cost-effectiveness of RALS versus CLS but may not be practical due to surgical preferences of surgeons and the extensive roll out of RALS.

## STRENGTHS AND LIMITATIONS OF THIS STUDY

⇒ Regression methods were used to account for differences in baseline characteristics between groups.
⇒ Cost estimates were derived from detailed microcosting of resource use.
⇒ Economic evaluation adhered to Consolidated Health Economic Evaluation Reporting Standards guidelines and used appropriate recommended methods.
⇒ Follow-up limited to 12 weeks post-surgery and as such does not consider potential longer-term effects.

## INTRODUCTION
### Background

Surgery used to treat early-stage gynaecological cancer has changed over the last 20 years through the introduction of minimal access surgical approaches. Surgeons have developed 'keyhole' (laparoscopic) techniques using 'straight stick' surgical instruments. This technique is termed conventional laparoscopic surgery (CLS). A number of studies have randomised patients with gynaecological cancer to CLS or open surgery. The Gynecologic Oncology Group (GOG) LAP2 study showed that patients receiving CLS had an improved safety profile, less antibiotic use and shorter hospital stay as well as higher quality-of-life scores (better physical functioning, better body image, less pain), earlier resumption of normal activities and earlier return to work over the 6 weeks following surgery,[1] compared with laparotomy in patients with endometrial cancer. A second trial (LACE study) showed quality of life improvements and an adverse event profile that favoured CLS compared with OH (open hysterectomy).[2] A Dutch study also confirmed that a laparoscopic approach provided benefits with respect to shorter hospital stay, less pain and

quicker resumption of daily activities compared with laparotomy.[1] CLS in gynaecological oncology has not been a panacea. The LAP2 and Dutch studies showed that despite the use of CLS, conversion rates varied from 10% to 26% and this rate increased to 57% in women with a body mass index (BMI) greater than 40 kg/m². Further, uptake of CLS in the UK has been slow and fragmented: for example, the ASTEC study (A Study in the Treatment of Endometrial Cancer, which recruited from 1998 to 2005) demonstrated that only 7% of patients being treated for endometrial cancer had a laparoscopic hysterectomy, while the remaining 93% had open surgery.[3] While the adverse events were lower in the CLS group compared with the OH group in ASTEC, a third of CLS patients in this study had their procedure converted to open surgery.

The introduction of robot assisted laparoscopic surgery (RALS) has facilitated a greater number of surgeons being able to offer minimal access surgery to their patients.[4] The combination of a wristed instrument design and three-dimensional high definition optics of the robot, allows the surgeon to dissect, remove and reconstruct tissues in a way that has not been possible.[5] Outcomes for RALS appear favourable to open surgery and there are a considerable number of publications suggesting that standard hospital performance indicators are as good for RALS as they are for other surgical techniques.[6–9] However, RALS brings with it a greater cost than CLS: operative times are longer, and equipment costs are greater for benign as well as oncological gynaecological procedures with no difference in key performance indicators.[4 10] A recent Cochrane review concluded '…we found limited evidence on the effectiveness and safety of RALS compared with CLS or open surgery for surgical procedures performed for gynaecological cancer; therefore, its use should be limited to clinical trials. Ongoing trials are likely to have an important impact on evidence related to the use of RALS in gynaecology…'.[11]

As a consequence, healthcare commissioners and policymakers in the UK have made RALS an area of investigation so that informed decisions can be made about the routine commissioning of robotics. For now, National Health Service (NHS) England has routinely commissioned RALS for the treatment of prostate cancer but for all other RALS services, robotic consumables are excluded from NHS tariff,[12 13] which is a process in which NHS providers in England receive healthcare funding. Simultaneous to the debate around healthcare costs, health and social care policymakers have focused on patient-reported outcome measures (PROMs). These are increasingly recognised as important in healthcare resource allocation decisions both globally and in the NHS.[14] NHS England analysis of PROMs collected following surgical procedures enables '…providers, commissioners and other stakeholders to make informed changes to the delivery of their services…'.[15] This data feeds into key NHS Outcomes Framework Indicators and the evidence of performance in these domains impacts on funding decisions across the healthcare system.[16]

There is little research which compares health-related quality of life between RALS and CLS. While Herling *et al* provided evidence of a speedy recovery in many domains following RALS, comparison of health-related quality-of-life recovery across different modes of surgery has not been explored.[17] Kumar *et al* attempted a comparison between open surgery and RALS, looking retrospectively at comparative health-related quality of life in endometrial cancer.[18] They found that RALS was associated with improved health-related quality of life compared with an open approach. The study however, did not compare RALS with CLS, and nor did it use validated health-related quality-of-life questionnaires or use recommended metrics for endometrial cancer.[19]

Any future decisions on provision of RALS in gynaecological cancer should be determined by informed decision-making of both the effectiveness and cost-effectiveness of this intervention compared with CLS. However, there is a paucity of data regarding comparative cost-effectiveness in this area.

## Objectives

The aims of this study were to compare the effectiveness in terms of health-related quality of life, and cost-effectiveness of RALS versus 'straight stick' CLS in women undergoing hysterectomy as part of their treatment for either suspected or proven gynaecological malignancy.

## METHODS
### Study design

This prospective comparative observational cohort study with embedded cost-effectiveness analysis recruited patients between August 2017 and February 2020 across 12 NHS Cancer Units and Centres in England. Patients referred for suspected gynaecological cancer or with a proven diagnosis of endometrial, cervical and ovarian cancer were given a patient information sheet and asked for consent to be part of this study. Demographic and anthropometric data was collected at baseline alongside relevant outcome measures and participants were followed-up at 2, 4 and 12 weeks post-procedure.

### Participant inclusion criteria

Inclusion criteria consisted of patients aged over 16 years who were referred with diagnosis (or suspected diagnosis) of gynaecological malignancy and about to undergo surgery RALS or CLS. Patients were also required to be fit for surgery, have the ability to complete the 6-minute walk test (6MWT) and they were also required to be having a hysterectomy as part of their surgical treatment to be eligible.

Sites were responsible for ensuring that patients had the capacity to give informed consent. A consent form which included an approved patient information sheet was provided in English to potential participants, with

an offer for an interpreter where necessary to ensure full informed consent from participants. In addition, an appropriately trained staff member at each site would provide a full explanation of the trial and additional trial requirements prior to trial entry of participants. An adequate amount of time was given to patients to consider the information and to discuss participation in the trial. Written informed consent was obtained before any trial specific procedures were conducted. Furthermore, parental consent was also required to be obtained for patients aged between 16 and 18.

Patients were excluded if surgery was only possible via the abdominal approach or if the midline laparotomy incision was anticipated to go beyond 4 cm above the umbilicus. Other exclusion criteria were surgery for a large ovarian tumour and patients undergoing radical cytoreductive surgery. Patients were also excluded if they were unable to complete outcome measures due to cognitive or language impairments disabling them from completing the quality-of-life questionnaires or lacked capacity to give consent.

### Public and patient involvement
Patient and public involvement were not included in this research project.

### Surgical procedures
Surgical procedures followed established practice for performing hysterectomy and bilateral salpingo-oophorectomy with/without pelvic lymph node dissection according to local cancer network guidelines for FIGO stage of disease (International Federation of Gynaecology and Obstetrics) as determined by availability of either surgically skilled robotic or laparoscopic surgeons.

### Primary outcome
The primary outcome for this study was the European Organisation for Research and Treatment of Cancer Quality of Life (EORTC QLQ C-30) health-related quality-of-life measure. The EORTC QLQ C-30 is a 30-item questionnaire assessing quality of life in patients with cancer. It contains five functional scales (physical, role, cognitive, emotional and social), three symptom scales (fatigue, pain and nausea and vomiting) and a global health and quality-of-life scale. The remaining single items assess additional symptoms commonly reported by patients with cancer (dyspnoea, appetite loss, sleep disturbance, constipation and diarrhoea), as well as the perceived financial impact of the disease and treatment. Higher scores indicate better function and heightened symptoms. The QLQ C-30 summary score is a validated measure for assessing quality of life for patients with cancer and it is presented as recommended by the EORTC Quality of Life Group.[20] The summary score is based on 27 items, the quality-of-life scale and the financial impact item are excluded.[20]

### Secondary outcomes
Secondary outcome measures included functional capacity at follow-up measured using the 6MWT,

health-related quality of life (EuroQol-5 Dimension (EQ-5D-5L) quality-adjusted life years (QALYs)), NHS costs and cost-effectiveness. The 6MWT is a submaximal test of aerobic capacity which requires the individual to walk as far as possible during 6 min around a 30 m course and the distance covered is recorded. It is a validated measure which has been recommended for use in patients with cancer.[21] Furthermore, the 6MWT has been applied in a clinical setting in a number of conditions as well as preoperatively and postoperatively.[22 23] The EQ-5D-5L comprises five domains each assessing a specific dimension of health-related quality of life (mobility, self-care, usual activities, pain and anxiety and depression) with five response levels ('no problems', 'slight problems' 'moderate problems', 'severe problems' and 'extreme problems').[24 25] Responses are converted into a weighted single utility score on a scale from 1 (perfect health) to 0 (death) in order to estimate QALYs.[24 25]

### Statistical methods
Reporting of study findings was informed by the Strengthening the Reporting of Observational studies guidelines.[26] Sample size was calculated for the primary outcome of this study, the health-related quality of life measured by the EORTC QLQ C-30. Minimum clinically important difference of the global health score of the EORTC QLQ C-30 was defined as a standardised mean difference of 0.3 SDs.[27] With 90% power (beta) and a 90% CI (alpha) it was anticipated that the study would require a total of 138 patients in each arm (with a maximum unequal allocation ratio of 2:1). Allowing for 25% attrition, this study aimed to recruit 173 patients in each cohort.

Patient characteristics and differences in surgical methods were compared using the appropriate statistical tests (t-test for continuous variables and $\chi^2$ test for categorical variables). The EORTC QLQ C-30 summary score, 6MWT and EQ-5D-5L utility were evaluated using a constrained baseline longitudinal regression model.[28] The equality at baseline constraint was imposed to counteract regression to the mean, whereby a group of patients with a worse average baseline score generally improved more than the group with better baseline scores, independent of any intervention effect.[29] This type of regression model also accounts for the hierarchical data structure and provides a principled method for dealing with missing outcome data, including missing baseline data with baseline as part of the outcome vector in this linear mixed model.[28] A Satterwhite correction of df was applied. The model adjusts for potential confounders identified as differences in baseline characteristics and surgical methods as fixed effects, alongside each individual patient modelled as a random effect. Parameters for interaction terms between time and treatment represent the estimated treatment effects at each follow-up. Treatment effects are reported with 95% CIs and p values. P values of less than 0.05 were considered statistically significant. All analyses were undertaken using Stata V.15.1.

## Economic evaluation

### Estimating the cost of surgery

The costing perspective was that of the UK NHS with all prices being in pounds sterling (£) using 2018–2019 prices. In order to increase precision and transferability into international contexts, the cost of each surgical procedure was micro-costed with resource use including equipment, staffing and consumables being collected from the study specific case report forms (CRF) and expert clinical guidance. General ward and intensive care unit (ICU) length of stay (LOS) following surgery until discharge were also collected from the study CRF. Following discharge, resource use was collected weekly for 12 weeks post-surgery using a Health Service Utilisation Questionnaire CRF. Follow-up resource use included primary care services, for example, general practice visits, district nurse visits as well as secondary care services, for example, inpatient stays, and outpatient visits. Serious adverse events (SAEs) from initial admission through to final follow-up were also recorded using a study specific CRF. Resource use is reported in online supplemental material S1.

Unit costs were derived from routine data sources[30 31] alongside study specific estimates. All reusable equipments were assigned a 'per use' cost. Where equipment had a short reusable lifespan (eg, 10 uses), the unit cost was divided by the number of uses to get a cost per use. Capital equipment such as laparoscopic stack systems and robotic components (as opposed to single or limited use surgical instrumentation) with prolonged lifespans were annuitised using a discount rate of 3.5%[32] to estimate yearly costs. These were then converted to a cost per use by assuming three surgical procedures per day across 254 days per year (this is the total number of working days Monday to Friday excluding public holidays in England). SAEs that occurred before discharge were included in the bed day costs. SAEs that required a hospital admission during follow-up were costed using the appropriate NHS tariff cost.[31] Unit prices are reported in online supplemental material S2.

### Quality of life

Quality of life was measured using EQ-5D-5L QALYs. To calculate a QALY, results from the EQ-5D-5L were converted into health state utilities at baseline, 2, 4, 12 weeks using a representative sample of the UK population.[33] These utility values were then used to estimate QALYS at 12 weeks using the area under the curve approach.[34]

### Estimating cost-effectiveness

A cost-effectiveness analysis at 12 weeks post-primary procedure was conducted to compare the incremental cost per QALY. The incremental cost-effectiveness ratio (ICER) was calculated as the difference in costs divided by the difference in QALYs between the two surgical groups using a bootstrap analysis with 1000 simulations. From this, a cost-effectiveness plane was produced, each quadrant indicating whether the surgical procedures were more or less expensive and more or less effective while also illustrating decision uncertainty.[35] For outcomes in which an intervention is both more costly and provided more QALYs can be further evaluated through consideration of willingness to pay, the additional cost willing to be paid in order to gain an additional QALY. To do this, a cost-effectiveness acceptability curve was produced which shows the number of bootstrapped iterations in which RALS was considered cost-effective over CLS across a range willingness to pay of threshold values (£0 to £50 000 per additional QALY gained). In England, National Institute for Health and Care Excellence (NICE) typically use willingness to pay threshold values of £20 000 to £30 000 per QALY.[36] These analyses were conducted using Stata V.15.1. The cost-effectiveness analysis was reported in accordance with the Consolidated Health Economic Evaluation Reporting Standards statement.[37]

One way sensitivity analysis was also conducted for the adjusted cost-effectiveness analysis to investigate how sensitive the results were to the assumptions for the annuitisation and per use cost of the capital equipment. This was conducted assuming different rates of daily usage (once, twice and four times) across 254 days per year.

## RESULTS

### Participants

A total of 275 patients recruited to this study, 29 patients were excluded as they did not have procedures recorded. Of the remaining 246 patients, 168 underwent RALS, 78 underwent CLS. Nine RALS patients and five CLS patients were excluded due to missing all outcome data. Overall, 159 RALS and 73 CLS were eligible for analysis. A flow diagram is presented in online supplemental material S3.

Baseline characteristics and surgical information of study participants are recorded in table 1. The average age, BMI, smoking status and number of prior surgeries were similar across both the RALS and CLS cohorts. However, there was a higher proportion of Caucasian patients in the CLS compared with the RALS cohort (n=68 (95.77%) versus n=129 (83.77%), p=0.011). The RALS cohort consisted of more patients with higher cancer grades (p=0.004) and higher cancer stages (p=0.012). There was no difference in EORTC or EQ-5D-5L health-related quality-of-life scores at baseline between both cohorts. The CLS cohort travelled more distance in the 6MWT at baseline compared with the RALS patients (386.54±109.71 vs 374.80±365.30, p=0.454).

### Intraoperative and length of stay

Very few patients had a lymph node dissection during CLS surgery compared with RALS (n=6 (8.22%) vs n=102 (64.14%), p=0.001). Of those who had a lymph node dissection in both surgeries, around half had a full lymph node dissection with the remainder having a sentinel procedure alone. The majority of both CLS and RALS procedures were simple hysterectomies (n=61

**Table 1**  Patient characteristics and surgical information

| Descriptor | RALS (n=159) | CLS (n=73) | P value |
|---|---|---|---|
| Demographics (baseline) | | | |
| Age (years) | 61.5±11.8 | 60.8±12.2 | 0.690 |
| BMI mean±SD | 31.5±8.0 | 30.6±7.6 | 0.763 |
| Smoking status | | | 0.849 |
| Non-smoker | 116 (72.96) | 55 (75.34) | |
| Ex-smoker | 6 (3.77) | 2 (2.74) | |
| Smoker | 26 (16.35) | 14 (19.18) | |
| Ethnicity | | | 0.011 |
| White | 129 (83.77) | 68 (93.15) | |
| Non-white | 25 (16.23) | 3 (4.11) | |
| Unknown | 11 (6.92) | 2 (2.74) | |
| Medical history (baseline) | | | |
| Number of prior surgeries | | | 0.476 |
| 0 | 129 (81.12) | 55 (75.34) | |
| 1 | 21 (15.72) | 15 (20.55) | |
| 2 | 8 (6.92) | 3 (4.11) | |
| 3 | 1 (0.63) | 0 (0) | |
| Unknown | 0 (0) | 0 (0) | |
| Histology | | | 0.145 |
| Adenocarcinoma | 15 (9.43) | 13 (17.81) | |
| Aden squamous | 2 (1.26) | 0 (0) | |
| Benign | 12 (7.55) | 10 (13.70) | |
| Carcinoma sarcoma | 5 (3.14) | 1 (1.37) | |
| Clear cell | 6 (3.77) | 1 (1.37) | |
| Endometrioid | 81 (50.94) | 36 (49.32) | |
| Sarcoma | 1 (0.63) | 1 (1.37) | |
| Serous | 24 (15.09) | 3 (4.11) | |
| Squamous | 6 (3.77) | 1 (1.37) | |
| Unknown | 7 (4.40) | 7 (9.59) | |
| Cancer grade | | | 0.004 |
| Benign | 12 (7.55) | 10 (13.70) | |
| 1 | 41 (25.79) | 29 (39.73) | |
| 2 | 27 (16.98) | 11 (15.07) | |
| 3 | 46 (28.93) | 7 (9.59) | |
| Unknown | 33 (20.75) | 16 (21.92) | |
| Cancer stage | | | 0.012 |
| No stage | 12 (7.55) | 10 (13.70) | |
| 1 | 110 (69.18) | 44 (60.27) | |
| 2 | 6 (3.77) | 8 (10.96) | |
| 3 | 15 (9.43) | 1 (1.37) | |
| 4 | 3 (1.89) | 0 (0) | |
| Unknown | 13 (8.18) | 10 (13.70) | |
| Outcomes (baseline) | | | |
| EORTC summary score | 86.29±11. 38 | 86.79±10.59 | 0.754 |
| EQ-5D-5L utility | 0.830±0.150 | 0.800±0.162 | 0.195 |

**Table 1** Continued

| Descriptor | RALS (n=159) | CLS (n=73) | P value |
|---|---|---|---|
| 6MWD (m) | 374.80±365.30 | 386.54±109.71 | 0.454 |
| Surgical methods and length of stay | | | |
| Lymph node dissection | | | <0.001 |
| Yes | 102 (64.14) | 6 (8.22) | |
| Full | 49 | 3 | |
| Sentinel | 53 | 3 | |
| No | 57 (35.85) | 67 (91.78) | |
| Type of procedure | | | 0.166 |
| Radical | 16 (10.06) | 12 (16.44) | |
| Simple | 143 (89.94) | 61 (83.56) | |
| Type of hysterectomy | | | 0.263 |
| With parametrium | 18 (11.32) | 12 (16.44) | |
| Without parametrium | 141 (88.68) | 60 (82.19) | |
| Theatre times | 183 (68) | 168 (93) | 0.179 |
| Ward days | 1.3 (2.1) | 1.8 (2.4) | 0.177 |
| ICU days | 0.08 (0.31) | 0.07 (0.34) | 0.774 |
| Total length of stay | 1.4 (2.1) | 1.9 (2.4) | 0.199 |

Data presented as n (%) or mean±SD.
CLS, conventional laparoscopic surgery; EORTC, European Organisation for Research and Treatment of Cancer; EQ-5D-5L, EuroQol-5 Dimension ; ICU, intensive care unit; 6MWT, 6-minute walk test; RALS, robotically assisted laparoscopic surgery.

(83.56%) and n=143 (89.94%), p=0.166). Radical hysterectomy made up a very small proportion of patients for both procedures. Hysterectomy without parametrium were n=60 (83.33%) and n=141 (88.68%) (p=0.263) in CLS and RALS, respectively. Theatre time was shorter on average for CLS (168±93 min) compared with the RALS (183±68 min) although this was not significant (p=0.179). LOS in terms of general ward bed days was higher for CLS relative to RALS (1.8±2.4 days vs 1.3±2.1 days, p=0.177). Very few patients required an ICU stay for either surgical procedure (mean LOS of 0.07±0.34 for CLS, 0.08±0.31 for RALS, p=0.774). As such, total LOS was slightly higher on average for CLS compared with the RALS (1.9±2.4 days vs 1.4±2.1 days, p=0.199). There were no deaths in either cohort up until first discharge date.

### Serious adverse events

There were 13 SAEs recorded during this study (online supplemental material S4), 6 occurred in CLS and 7 in RALS. For CLS, three occurred following the procedure but before discharge with another three were detected over the first 2 weeks of follow-up. For RALS, there were six SAEs prior to discharge, and one SAE in week 2 of follow-up.

### Primary outcome

The crude and adjusted treatment effects for all outcomes are presented in table 2. Mean EORTC health-related quality of life at the 2-week follow-up compared with baseline fell for both RALS and CLS (−15.22±13.82,

−15.97±15.28). The adjusted between-group difference (treatment effect) in EORTC from baseline to week 2 post-surgery was higher for RALS (1.6 (95% CI −2.8 to 6.1)). The mean EORTC at week 4 for both surgeries were also lower than baseline (−5.95±13.31 for RALS and −4.49±13.65 for CLS). The adjusted between group difference in the EORTC score from baseline to week 4 was lower for RALS (−0.3 (95% CI −4.8 to 4.1)). By 12 weeks follow-up, the EORTC score compared with baseline, had improved in both RALS (0.85±11.36) and CLS (2.09±11.45), respectively, with an adjusted between group difference of 0.4 (95% CI −4.1 to 4.9) in favour of RALS.

### Secondary outcomes

For both RALS and CLS, EQ-5D utility scores were lower in weeks 2 and 4 compared with baseline but were higher by week 12. The distance travelled in the 6MWT was lower for RALS and CLS in week 2 compared with baseline but was higher in weeks 4 and 12. However the adjusted analysis for both outcomes showed no statistically significant treatment effect.

### Costs

Complete data was obtained for each participant regarding procedure costs and costs up until discharge. Summary costs of procedure and follow-up for both RALS and CLS are presented in online supplemental material S5. Procedure costs included costs relating to staff, theatre time, equipment and LOS from admission

**Table 2** Crude change in outcomes from baseline at different follow-up points and estimated adjusted treatment effects with 95% CI

| Time-point | Number of participants | | Crude change from baseline * | | Estimated treatment effect (group difference) in mean change† with 95% CI | P value |
|---|---|---|---|---|---|---|
| | RALS | CLS | RALS | CLS | | |
| Primary outcome | | | | | | |
| EORTC score | | | | | | |
| 2-week follow-up | 114 | 55 | -15.22±13.82 | -15.97±15.28 | 1.6 (–2.8 to 6.1) | 0.469 |
| 4-week follow-up | 112 | 55 | -5.95±13.31 | -4.49±13.65 | -0.3 (–4.8 to 4.1) | 0.884 |
| 12-week follow-up | 103 | 51 | 0.85±11.36 | 2.09±11.45 | 0.4 (–4.1 to 4.9) | 0.861 |
| Secondary outcomes | | | | | | |
| 6MWD (m) | | | | | | |
| 2-week follow-up | 86 | 52 | -12.31±84.05 | -13.71±104.04 | 18.1 (–11.0 to 47.2) | 0.223 |
| 4-week follow-up | 102 | 49 | 15.46±71.03 | 31.08±86.96 | -3.4 (–33.2 to 26.4) | 0.823 |
| 12-week follow-up | 82 | 35 | 27.33±86.94 | 32.48±74.11 | 3.4 (–27.6 to 34.4) | 0.830 |
| EQ-5D-5L utility | | | | | | |
| 2-week follow-up | 115 | 57 | -0.11±0.20 | -0.09±0.21 | 0.036 (–0.023 to 0.095) | 0.227 |
| 4-week follow-up | 114 | 55 | -0.06±0.16 | -0.01±0.20 | 0.010 (–0.050 to 0.069) | 0.745 |
| 12-week follow-up | 107 | 50 | 0.01±0.16 | 0.02±0.20 | -0.002 (–0.061 to 0.058) | 0.959 |

*Crude change from baseline given as mean change±SD of this change.
†Estimated treatment effect for all outcomes are the between group differences in the change from baseline derived from the mixed effects model adjusting for lymph node dissection and cancer stage as fixed effects.
CLS, conventional laparoscopic surgery; EORTC, European Organisation for Research and Treatment of Cancer Quality of Life; EQ-5D-5L, EuroQol-5 Dimension ; 6MWD, 6-minute walk distance; RALS, robotically assisted laparoscopic surgery.

to discharge. The mean procedure cost was higher for RALS compared with CLS (£4758±972 vs £3741±1350, p<0.001). This was due to RALS having a higher mean staff costs (£1851±688 vs, £1703±940, p=0.231), theatre costs (£765±284 vs £704±388, p=0.231) and equipment costs (£2 134 vs £1325). However, the mean cost for total LOS following the procedure was lower for RALS (£704±1002 vs £851±1159, p=0.352). Overall, the total mean cost from admission to discharge including the procedure costs were higher for RALS compared with CLS (£5462±1472 vs £4592±2072, p=0.002).

Follow-up resource was collected up to 12 weeks post-discharge. In weeks 1–3 follow-up, post-discharge mean costs were higher for CLS relative to RALS, likely due to higher overnight stays in hospital. Thereafter, mean costs incurred in each week were similar between the two groups. Costs in week 12 were higher for RALS due to secondary care costs.

## Cost-effectiveness analysis

Table 3 reports the outcomes of the constrained base-line longitudinal regression model unadjusted (no

**Table 3** Results for the cost-effectiveness models

| | Unadjusted analysis | Adjusted analysis* |
|---|---|---|
| Cost difference† (£) | 556 (–314 to 1315) | -188 (–1321 to 827) |
| QALYS difference† | 0.0024 (–0.0005 to 0.0057) | 0.0024 (–0.0010 to 0.0057) |
| Non-parametric ICER (£ per QALY) | 231 667 | Dominated |
| Willingness to pay (£20 000 threshold)‡ | 0.117 | 0.657 |
| Willingness to pay (£25 000 threshold)‡ | 0.126 | 0.664 |
| Willingness to pay (£30 000 threshold)‡ | 0.133 | 0.671 |

Differences are shown as RALS—CLS.
*Models adjust for lymph node dissections and cancer stage.
†Reported as mean (95% CI).
‡Proportion of bootstrapped iterations in which RALS is cost-effective.
CLS, conventional laparoscopic surgery; ICER, incremental cost-effectiveness ratio; QALY, quality-adjusted life year; RALS, robotically assisted laparoscopic surgery.

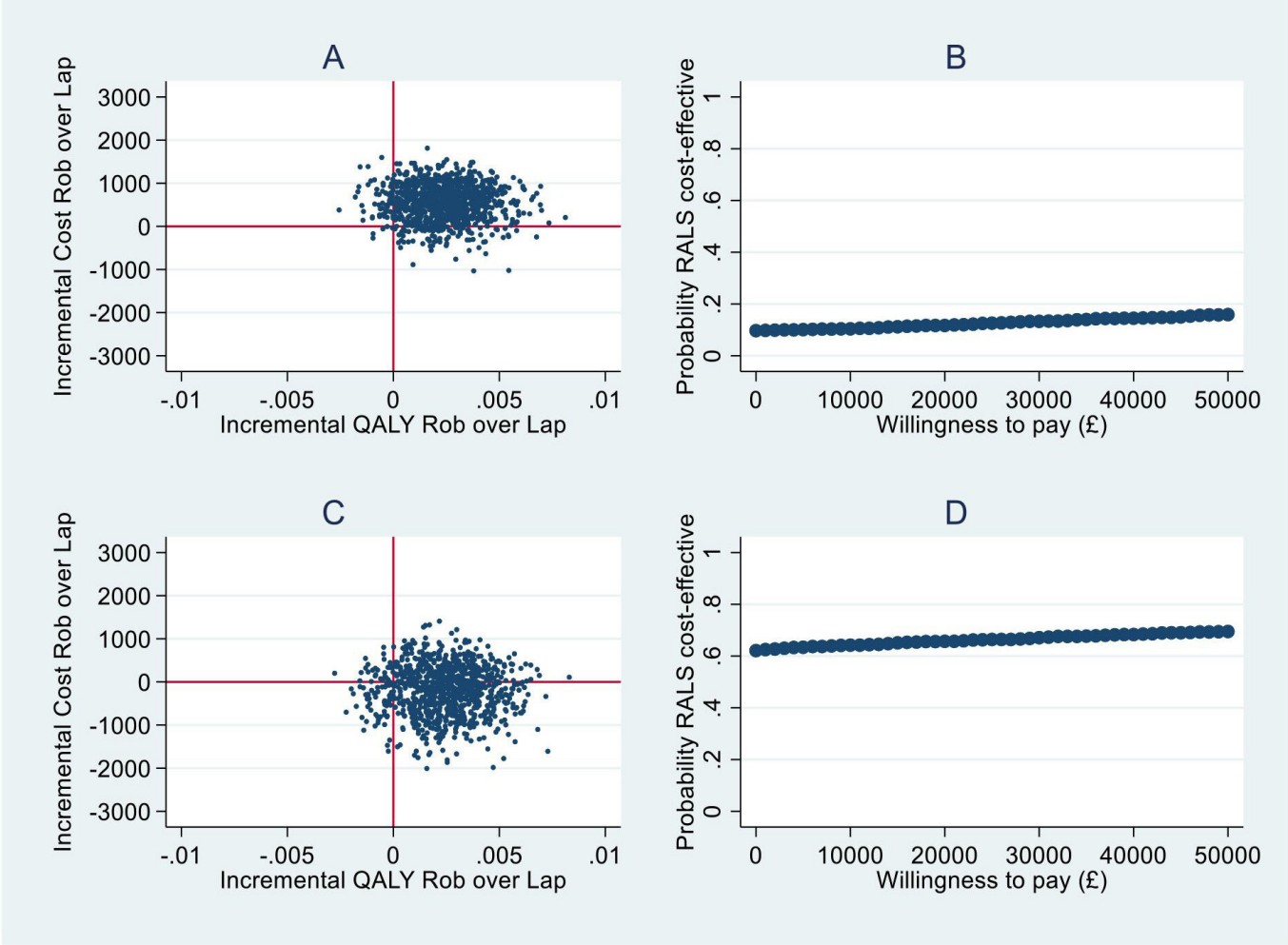

**Figure 1** (A) Cost-effectiveness plane for unadjusted analysis. (B) Willingness to pay for unadjusted analysis. (C) Cost-effectiveness plane for adjusted analysis. (D) Willingness to pay for adjusted analysis. QALY, quality-adjusted life years; RALS, robot assisted laparoscopic surgery.

confounder) and adjusted cost-effectiveness analyses with figure 1 presenting the cost-effectiveness acceptability curves and the willingness to pay.

For the unadjusted cost-effectiveness analysis, including follow-up over 12 weeks, the RALS cohort had a mean total cost difference of £556 (95% CI –£314 to £1315) compared with CLS and a mean QALY difference of 0.0024 (95% CI –0.00051 to 0.0057). This resulted in a non-parametric ICER of £231 667 per QALY. At a willingness to pay threshold of £20 000 per QALY, 11.7% of the bootstrapped iterations resulted in RALS surgery being cost-effective, rising to 13.3% of iterations being cost-effective at £30 000 per QALY.

When adjusting for lymph node dissections and cancer stage, over 12 weeks, RALS had a mean cost difference of –£188 (95% CI –£1321 to £827) compared with CLS and a mean QALY difference of 0.0024 (95% CI –0.0008 to 0.0057). This resulted in a non-parametric ICER being dominant. At a willingness to pay threshold of £20 000 per QALY, 65.7% of the bootstrapped iterations resulted in RALS being cost-effective, rising slightly higher to

67.1% of iterations being cost-effective at a £30 000 per QALY threshold.

### Sensitivity analysis
The results of the one-way sensitivity analyses are presented in table 4 with the cost-effectiveness plane, and willingness to pay figures presented online supplemental material S6. RALS surgery had a higher average cost compared with CLS when assuming only one procedure per day (mean cost difference £430 (95% CI –£703 to £1445)), with a resulting non-parametric ICER of £179 167 per QALY. When assuming only one procedure per day, only 23.8% of the bootstrapped iterations are cost-effective with a willingness to pay of £20 000 per QALY. When assuming two or four procedures per day, the cost of the robot is lower than CLS (mean cost difference: –£34 (95% CI –£1167 to £981) and –£266 (–£1399 to £749)), respectively. The non-parametric ICER for both scenarios were dominant in both cases, suggesting that RALS surgery provides more QALYs and has a lower cost compared with CLS. Assuming two procedures per day,

**Table 4** Results of the one-way sensitivity analysis of the adjusted cost-effectiveness model, varying the number of procedures performed per day

|  | One procedure per day* | Two procedures per day* | Four procedures per day* |
|---|---|---|---|
| Cost difference† (£) | 430 (–703 to 1445) | -34 (–1167 to 981) | -266 (–1399 to 749) |
| QALYS difference† | 0.0024 (–0.0010 to 0.0057) | 0.0024 (–0.0010 to 0.0057) | 0.0024 (–0.0010 to 0.0057) |
| Non-parametric ICER (£ per QALY) | 179 167 | Dominated | Dominated |
| Willingness to pay (£20 000 threshold)‡ | 0.238 | 0.536 | 0.700 |
| Willingness to pay (£25 000 threshold)‡ | 0.246 | 0.553 | 0.703 |
| Willingness to pay (£30 000 threshold)‡ | 0.256 | 0.569 | 0.714 |

Differences are shown as RALS—CLS.
CLS conventional laparoscopic surgery, RALS robotically assisted laparoscopic surgery.
*Models adjust for lymph node dissections and cancer stage.
†Reported as mean (95% CI).
‡Proportion of bootstrapped iterations in which RALS is cost-effective.
CLS, conventional laparoscopic surgery; ICER, incremental cost-effectiveness ratio; QALY, quality-adjusted life years; RALS, robot assisted laparoscopic surgery.

RALS was cost-effective for 53.6% of bootstrapped iterations, and when assuming four procedures per day RALS was cost-effective for 71.4% of the bootstrapped iterations at a willingness to pay of £20 000 per QALY.

## DISCUSSION

The delivery of surgical care in gynaecological cancer in the UK NHS continues to evolve. Minimal access surgery remains the mainstay for treating patients with endometrial cancer. Robotic surgery continues to be rolled out across NHS organisations but there is no national programme to coordinate this, and the availability of robotic surgery is dependent on local trust finances and agreements with local commissioning groups.[38] Thus, the type of surgical procedure a patient is offered to treat gynaecological malignancy is dependent on their postcode and what is available in their local NHS hospital.

This pragmatic prospective comparative observational cohort study compares outcomes and cost-effectiveness for patients undergoing CLS or robotically assisted laparoscopic surgery reflecting real world care pathways. The results in terms of impact on functional outcomes and health-related quality of life showed very little difference between the two surgical methods and are in keeping with previous findings.[39 40] Therefore, the results of the cost-effectiveness analyses were largely driven by costs, including the cost of procurement and maintenance of the robotic device and subsequent changes in resource use after surgery.

The unadjusted cost-effectiveness analysis in our study suggested that RALS was not cost-effective using the NICE threshold, of £30 000/€34 000.[41] However, as with any observational study there were differences in characteristics of the patients in each of the surgical groups which led to confounding and bias in the unadjusted analysis. The characteristics in question were related to higher cancer stages and a higher number of lymph node evaluations in the RALS group. Explanation for these

differences is difficult to attribute as one would expect patients to be treated by the same protocol according to the national guidelines which exist in England to direct practice. However, our results suggest that other factors come into play when undertaking surgery, for example, as BMI rises surgical factors may influence what technically can be achieved between the two platforms.

Once these adjustments were made to the analysis, the probability of RALS being cost-effective was much higher. The difference in cost-effectiveness between the unadjusted and adjusted base case analysis was driven by cost differences. One possible explanation for this difference may be that the higher cancer stages and increased number of lymph node evaluations in the RALS cohort resulted in longer theatre times and subsequently higher theatre and staff costs.

Despite national guidelines[42] on best practice for lymph node evaluation, current practice varies across the UK dependent on local resources and surgical paradigm. Further, we noted that some hysterectomies had been classified as 'radical', yet they had not had parametrial or lymph node dissections. Further analysis (data not shown) revealed that these patients had surgery for endometrial cancer rather than cervical cancer. The explanation for this lay in the way in that NHS Trusts coded their surgical procedures on the basis of commissioning because this had implications for payments to Trusts on procedures performed. For example, some operations were coded as a modified radical hysterectomy if there was any element of ureterolysis. This meant that some endometrial cancers were described as being 'modified radical hysterectomy' but lymph node dissections may not have been indicated, as per local guidelines. The authors confirmed in those centres where this discrepancy was identified that all gynae oncologists would perform full bilateral pelvic lymphadenectomies for those patients having a radical hysterectomy for cervical cancer.

To our knowledge this study is the first to conduct a multicentre full cost-effectiveness evaluation comparing RALS versus CLS for hysterectomies following gynaecological cancer. Other studies in this area only focus on costs. An older prospective single centre study comparing costs of RALS versus CLS was conducted by Sarlos et al.[43] This study collected prospective data from June 2007 to May 2009 of 40 RALS patients following the introduction of robotic surgery in a single centre and compared this with a matched set of 40 CLS patients from retrospective records. Patients were matched 1:1 according to surgeon, uterus weight, BMI and age. Unlike the findings of our adjusted analysis, Sarlos et al[43] estimated a higher cost of RALS £4066.84 compared with CLS £2150.76. However, we note that although the costs of Sarlos et al[43] are micro-costed, they do not include the cost of the robotic device. Furthermore, one potential driver of higher costs of RALS in their study could be due to surgeon inexperience. The RALS centres in our study include surgeons who were experienced in performing RALS and likely represents a truer reflection of practice today. As they conducted a matched case control study, Sarlos et al[43] do not present adjusted costs. However, we argue that there are potential limitations in the matching used as no consideration is given to patient comorbidities, which have been identified as drivers of cost.[44]

Another study by Ind et al[45] investigated the costs of RALS compared CLS in a prospective cohort study of a single centre in England, where surgeons and surgical teams had prior experience with robotic surgery. It was noted that the robotic device was donated by a charity. Although the robotic device was donated to the centre, the micro-costing conducted by Ind et al[45] also included a depreciated per use cost of the purchase of the robotic device. The results of Ind et al are similar to our own study, with the 30-day cost of RALS estimated to be lower at £8481 compared with CLS at £9979. However, it should be noted that the cost estimates of both interventions are much higher than in our own study due to Ind et al[45] reporting much higher theatre costs, LOS and high dependency use. For example, Ind et al[45] report a median LOS of 3 days for CLS and 2 days for RALS with 80.5% of CLS and 70.8% of RALS patients requiring high-dependency postoperative care. This differs from our own findings in which median LOS for CLS and robotic surgery was 1 day each, with very few patients in either arm requiring stay in ICU. Furthermore, length of time for surgery reported by Ind et al[45] was higher for both interventions compared with our study. These differences may be due to factors such as changes in practice, advances in both RALS and CLS, or due to small sample size of Ind et al.[45]

A larger, more recent retrospective cohort study was conducted by Moss et al[44] comparing costs of different routes of surgery for the management of endometrial cancer with up to 1-year follow-up using NHS England Hospital Episode Statistics (HES) data between 2011 and 2018. The unadjusted cost at 90 days in the RALS cohort was on average £246 more expensive compared with CLS. However, the patients undergoing RALS had higher Charlson Comorbidity Index scores and an increased number were obese compared with CLS. When controlling for the differences between the cohorts, Moss et al[44] estimated a reduced cost difference at 90 days with RALS being only £89 more expensive. These results are similar to our own findings with the cost difference identified by Moss et al[44] situated within the 95% CI of the cost difference in our adjusted analysis –£188 95% CI (−1321 to 827). However, there are still some limitations to their study compared with our own. First, Moss et al[44] used Healthcare Resource Group (HRG) tariff to estimate costs rather than detailed micro-costing. Furthermore, unlike our detailed micro-costing the HRG tariff estimates do not include capital costs or maintenance costs of the robotic devices, which as our study shows is a large and important component of surgical costs with robotic surgery. As a prospective multicentre cohort study, our study uses a much more rigorous inclusion criteria compared with Moss et al[44] and data collection of resource use may be more accurate in our study as we do not rely on HES data and also capture primary care resource use.

The sensitivity analysis conducted as part of our study identified that results of the cost-effectiveness analysis were highly sensitive to the volume of procedures performed. The finding that increasing the usage of the robotic devices increases the probability of RALS being cost-effective is in keeping with cost-effectiveness analyses comparing robotic surgery in other pathologies.[46] We therefore argue that the relationship between usage and cost-effectiveness of RALS should stimulate centres using robotic devices to maximise the volume of cases that are carried out using this equipment.

Acquisition of surgical robots has been managed in England to date by individual hospital trusts and a reliance on charitable donations with no centralised strategy. This may have unintended consequences for health service provision. First, trusts may feel obligated to prioritise treatment of diseases on which the donating charity is focused, even where this may not be cost-effective. Furthermore, we question the long-term viability of relying on charitable donations, particularly when the robotic devices become obsolete and reach the end of their life span and need to be replaced. Over-reliance on charitable donations may not be sustainable into the future as RALS moves to becoming more mainstream.

Lam et al examined the adoption of surgical robots through a nationwide Freedom of Information Request to all acute NHS Trusts in England[38] and confirmed our argument regarding increasing the usage of the robotic equipment. Specifically, their results suggested that despite the number of robotic procedures increasing, national accessibility to robotic services and case volumes were highly variable with many trusts having low volume use representing missed efficiency savings for the NHS. We therefore argue that policies regarding the deployment and use of robotic surgical equipment are needed

at national, regional and/or hospital level depending on the configuration of the healthcare system. One possible solution to this challenge in publicly funded national health systems such as the UK NHS is the centralisation of robotic centres into 'innovation hubs' that can establish best practice, clinical evidence and patient volumes.[38] In the absence of centralised initiatives, hospital operational policies should aim to procure robotic equipment over a range of surgical specialties parallel to a drive to undertake procedures in extended time slots to aid cost-effectiveness.

## Limitations

We acknowledge the limitations of this study as a pragmatic prospective cohort study. This was necessary because of the current geography and delivery of RALS and CLS and the hub and spoke structure of cancer services in the UK. The fact that the two populations of patients differed in simple factors such as grade and stage of cancer has impacted the generalisability of this data. Although this study contained 12-week follow-up, in which there was no mortality recorded, it should be noted that stage differences may impact longer-term mortality. Future studies which consider longer-term outcomes may need to consider the impact that stage differences may have on outcomes. Of some reassurance in terms of justifying robotic costs in endometrial cancer is the finding that the higher stage and grade patient cohort showed a swing in cost-effectiveness favouring robotics despite worse prognostic factors. A randomised multicentre prospective trial would circumvent this limitation and provide data with greater generalisability to organisations wanting to establish the cost-effectiveness of robotics ahead of implementation of a programme in their institution. However, design of such a trial would need significant and careful consideration to overcome the difficulties of acceptability to patients and surgeons alike, nationally and internationally, so that recruitment and ownership of the trial made it successful. In addition, the expansion of RALS across the NHS is advancing and could make it difficult to recruit enough CLS participants in any future trial, and indeed may have contributed towards the under recruitment of CLS patients in this study. Identifying cost-effectiveness through the gold standard randomised controlled trial (RCT approach may not be feasible and as such this study is of great value in adding to the literature regarding the cost-effectiveness comparison of RALS and CLS and helping to inform decision-makers on efficient use of resources.

Although this study recruited a sufficient number of RALS patients, it struggled to recruit CLS patients and as such results of the primary analysis are underpowered. Attrition was lower than anticipated, however CLS still fell short of requirements. It should be noted however, that the cost-effectiveness analysis collected 'real world evidence' of costs and utilities for patients undergoing CLS and RALS, which will be informative and of great value to healthcare funders. Furthermore,

cost-effectiveness analysis has long been recognised as being underpowered as a secondary outcome in many studies.[47] The techniques used in this study follow recommended guidelines and focus on estimating the cost and effect differences alongside the likelihood of and intervention being cost-effective, rather than hypothesis testing, with consideration given to uncertainty around point estimates and comparisons.[48]

This study did not include any cases which required conversion to open surgery. Future cost-effectiveness studies would benefit by accounting for these cases as they may impact cost-effectiveness results. A meta-analysis of robotic surgery[49] suggests that RALS for endometrial cancer offers lower conversion rates than CLS and as such it may be inferred that accounting for conversion cases may further increase the cost-effectiveness of RALS versus CLS.

A further limitation of this study is that confounding adjustment was managed within a regression model by including control variables. This required a well specified model to reduce residual confounding and consequent bias. Propensity scoring is an alternative and increasingly popular method of controlling potential confounding in observational studies that compare the effectiveness of healthcare interventions.[50 51] Propensity matching is typically used in cohort studies and involves fitting regression models to predict treatment groups based on selected characteristics.[51] However, there is evidence to suggest that there is equivalence between confounding adjustment to various propensity score-based approaches.[52 53]

## Conclusion

This study identified no differences in health-related quality of life or functional outcomes between patients undergoing RALS or CLS for treatment of confirmed and suspected gynaecological endometrial and cervical cancers. However, this study has shown that there is evidence to suggest that RALS may be cost-effective compared with CLS, but results are highly sensitive to assumptions around the usage, and subsequent per use cost, of the robotic hardware. As robotic surgery becomes more mainstream, it is of vital importance that usage of robots are optimised to ensure efficient resource use. Further investigation is needed in this area is needed and a non-inferiority RCT would be of benefit in decision-makers with regards to cost-effectiveness, however this may not be practical or achievable in the NHS.

**Author affiliations**
[1]Department of Nursing, Midwifery and Health, Northumbria University, Newcastle upon Tyne, UK
[2]Department of Gynaecology Oncology, University Hospitals Bristol and Weston NHS Foundation Trust, Bristol, UK
[3]University of Bristol, Bristol, UK
[4]Department of Gynaecological Oncology, Royal Marsden NHS Foundation Trust, London, UK
[5]Department of Gynaecological Oncology, University Hospitals Dorset NHS Foundation Trust, Poole, UK
[6]Department of Gynaecological Oncology, Coventry and Warwickshire Hospital, Coventry, UK

**Acknowledgements** Sarah Kiddell (formerly Clinical Trial Practitioner for South Tees Hospitals NHS Foundation Trust) was the Trial Manager with primary responsibility for the day-to-day running of the study and reporting duties. Richard Hodgson (formerly Research Data Officer for South Tees Hospitals NHS Foundation Trust), followed by Lesley Harris (formerly Research Data Officer for South Tees Hospitals NHS Foundation Trust) were the Data Support Officers checking data completeness and accuracy. Julie Rowbotham (formerly Research Manager for South Tees Hospitals NHS Foundation Trust) acted as Sponsor Representative for the research study, followed by Joe Millar (Research Governance Manager for South Tees Hospitals NHS Foundation Trust). Both were responsible for overseeing the delivery of sponsorship responsibilities and providing research governance advice throughout. The following contacts are all acknowledged for their role as Principal Investigator for each of the listed research sites: Simon Butler-Manuel, Royal Surrey NHS Foundation Trust. Allan Gillespie, The Sheffield Teaching Hospitals NHS Foundation Trust. Pierre Martin-Hirsch, Lancashire Teaching Hospitals NHS Foundation Trust. Richard Edmondson, Manchester University NHS Foundation Trust. Mike Smith, The Christie NHS Foundation Trust. David Milliken, Somerset NHS Foundation Trust. Partha Sengupta, Country Durham & Darlington NHS Foundation Trust. Tony Challoub, Newcastle upon Tyne Hospitals NHS Foundation Trust.

**Contributors** The study was conceptualisation by JG, PM, CN, MN, JL and JT. Data was curated by AM, DS and SM with formal analysis being conducted by AM, DS, JG, PM and SM. Funding was acquired by JG and JT. All authors were involved in the investigation and interpretation of data with methodology provided by AM, DS, JG, SM and PM. Original draft was written by AM and JG with all authors contributing to the review and editing. Final approval was obtained from all authors before submission. AM is the guarantor and attests that all listed authors meet authorship criteria and that no others meeting the criteria have been omitted.

**Funding** The study was supported by Intuitive through a £93 850 funding grant and provision of a secure data management web service. Grant/award number: N/A. However, the study was managed and conducted by a wholly independent project team. The grant funding paid for identified project costs such as project management staff support, statistician and health economist support, equipment, supplies and hospital staff support.

**Competing interests** All authors have completed the ICMHE uniform disclosure form at http://www.icmje.org/disclosure-of-interest/ and declare: all authors had financial support from Intuitive for the submitted work. JL has received renumeration for time given as a proctor at intuitive, MN has received payment from intuitive for consultation and also for time given as a proctor and for development of educational lectures, in addition MN is President of the British & Irish Association of Robotic Gynaecological Surgeons (BIARGS); no other relationships or activities that could appear to have influenced the submitted work.

**Patient and public involvement** Patients and/or the public were not involved in the design, or conduct, or reporting, or dissemination plans of this research.

**Patient consent for publication** Not applicable.

**Ethics approval** This NIHR Portfolio research study fully conformed to the ethical norms and standards in the Declaration of Helsinki. The study received the appropriate regulatory approvals in the UK before proceeding having been approved by an NHS Research Ethics Committee and registered with each participating NHS Trust as per their local research department's process. Ethical approval was obtained from West Midlands Solihull Research Ethics Committee (NG1 6FSIRAS Ref 221315 - REC Ref - 17/WM/0056 R&D Approval date 22/06/2017). Furthermore, South Tees Hospitals NHS Foundation Trust as sponsor and trial coordinating organisation carried out several duties (such as provision of Site Initiation Visits and check-in calls, data validation reports, monitoring visits) to promote high quality trial conduct and adherence to Good Clinical Practice.

**Provenance and peer review** Not commissioned; externally peer reviewed.

**Data availability statement** Data are available upon reasonable request. The complete research data supporting the findings of this paper were all stored in a web research database maintained by the funder Intuitive, which has now been data locked and exported as an excel record held by both Intuitive and the project team. Data is available on request due to privacy/ethical restrictions.

**ORCID iD**
Andrew McCarthy http://orcid.org/0000-0002-3385-6302

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
