## [Reviewer comments · BMJ Open]

ARTICLE DETAILS

TITLE (PROVISIONAL)	Robotic and Laparoscopic Gynaecological Surgery: A Prospective Multi-Centre Observational Cohort Study and Economic Evaluation in England
AUTHORS	McCarthy, Andrew; Samarakoon, Dilupa; Gray, Joanne; Mcmeekin, Peter; McCarthy, Stephen; Newton, Claire; Nobbenhuis, Marielle; Lippiatt, Jonathan; Twigg, Jeremy

VERSION 1 – REVIEW

REVIEWER	Simianu, Vlad V Virginia Mason Medical Center
REVIEW RETURNED	10-Jun-2023

GENERAL COMMENTS	The authors of this paper put forth a well thought out, thorough economic analysis of the cost-effectiveness of robotics vs laparoscopy when performing hysterectomies for gynecologic cancer in the United Kingdom. Given that cost can make certain technologies and treatments cost prohibitive for a variety of reasons, a study such as this adds to the overall fund of knowledge as providers and hospital systems attempt to decide how to best allocate resources to treat patients. As a cost-effectiveness evaluation, there are a few critical components that need to be addressed. Major Concerns: 1. The baseline differences in lymph node dissection bw approaches is striking – can authors comment on that?2. Perhaps as it relates to above point, would expect some role for stage differences to impact survival, and therefore time alive beyond 12 weeks3. The authors conduct the analysis from a limited health system perspective – it would benefit from using the societal perspective, incorporating things like patients return to work, and QOL/survival beyond 12 weeks.4. It's a little surprising to see that the sensitivity analysis didn't include variation on PROMs or instrument costs (which have ranges in the provided data), especially since this is observational data. This would be a valuable addition, either as one-way or multi-way sens analyses: Sanders GD, Neumann PJ, Basu A, Brock DW, Feeny D, Krahn M, Kuntz KM, Meltzer DO, Owens DK, Prosser LA, Salomon JA, Sculpher MJ, Trikalinos TA, Russell LB, Siegel JE, Ganiats TG. Recommendations for Conduct, Methodological Practices, and Reporting of Cost-effectiveness Analyses: Second Panel on Cost-Effectiveness in Health and Medicine. JAMA. 2016 Sep 13;316(10):1093-103. doi: 10.1001/jama.2016.12195. Erratum in: JAMA. 2016 Nov 8;316(18):1924. PMID: 27623463.
---

	Minor Concerns:  1. In the introduction the authors reference the ASTEC study, stating that a 3rd of conventional laparoscopic surgery patients in that study had the procedures converted to open surgery. I believe the authors excluded converted to open operations, but it would be good to include these rates. It would be interesting if the authors could comment on if there is any evidence that these conversions could have been prevented with the use of robotics given their finding that the robotic cases in their analysis were patients with cancers at higher grade and stage. 2. In the introduction there is a reference that “robotic consumables are excluded from tariff”. Given the international readership of this journal, recommend expanding the explanation as readers from outside the UK may not understand what this means for robotics usage. 3. Remove “Functional Capacity” from the objective aims as the paper does not discuss this that much and could take away from the strength of the cost-effectiveness analysis. 4. Table 1 and Supplement 4 are the same. Take our supplement 4 5. Add a description of what “Willingness to pay” means. This was unclear and confusing, taking away from the strength of the analysis 6. What is the National Institute for Health and care excellence threshold (again, for international readers)
--	---

REVIEWER	Johannesson, Ulrika Karolinska Institute
REVIEW RETURNED	16-Jul-2023

GENERAL COMMENTS	This is a well written manuscript considering the effectiveness and cost effectiveness of robot-assisted laparoscopic surgery (RALS) versus conventional “straight stick” laparoscopic surgery (CLS) in women undergoing hysterectomy with a gynaecological malignancy indication. I have the following comments and questions;  1. Did you perform a power analysis? Do you have enough patients in the study to determine differences in HRQOL? What is the base population for this cohort (how many procedures in total during this time in the 12 NHS centres)? Please elaborate since this might improve the full picture and validity. 2. Please clarify “effectiveness” in the abstract. 3. Can you please motivate the choice of questionnaires? An explanation might further improve your manuscript. 4. HRQOL-please revise spelling of HRQOL, QOL etc throughout manuscript. 5. The dropouts, what happened to them? Results section page 12, line 12-14 “A total of 275 patients recruited to this study, 29 patients were excluded as they did not have procedures recorded”. Please specify/define what this drop-out means. In the abstract it says “298 patients recruited with 159 RALS, 73 CLS eligible for analysis” Explain the difference in figures, 298 vs 275. 6. Could a definitive RCT with embedded full economic evaluation be performed elsewhere in the world? Suggestions on how to further analyze and evaluate your results. 7. Introduction Page 5 Line 50-53 Abstract missing comma `robot allows the surgeon to dissect remove and reconstruct tissues in a way that has not been possible`
--

	In conclusion, I find this article highly relevant and with these minor revisions suitable for publication.
--	---

VERSION 1 – AUTHOR RESPONSE

Response to reviewer 1:

Major Concerns:

1. The baseline differences in lymph node dissection bw approaches is striking – can authors comment on that?

The differences in lymph node dissection approaches are difficult to attribute. We would expect patients to be treated by the same protocol according to national guidelines. However, our results suggest that there may be other factors (e.g. BMI) that come in to play when undertaking surgery. We have added the following to the discussion section in order to further address these differences within the manuscript.

“...Explanation for these differences is difficult to attribute as one would expect patients to be treated by the same protocol according to the national guidelines which exist in Britain to direct practice. However, our results suggest that other factors come in to play when undertaking surgery, for example as body mass index rises surgical factors may influence what technically can be achieved between the two platforms...”

2. Perhaps as it relates to above point, would expect some role for stage differences to impact survival, and therefore time alive beyond 12 weeks

We agree that stage differences may impact longer-term mortality however this study focussed on the initial recovery from surgical intervention and followed patients up to 12-weeks. It should be noted that we did not identify any mortality within the 12-weeks within our patient cohort. The differences in stages between cohorts was unexpected in this cohort study, these differences may need to be considered in the design of any future studies in this area. This may be particularly important if considering a longer-term follow-up period.

However, in the discussion we compare similar studies to our own which also investigate costs differences between RALS and CLS and note that follow-up was limited to less than 90 days, with the exception being Moss et al 2021 which relied on routinely collected health episode statistics data as a secondary data source to identify costs up to 1 year only.

We have addressed this within the manuscript with the addition of the following in the limitations section.

“Although this study contained 12-week follow-up, in which there was no mortality recorded, it should be noted that stage differences may impact longer term mortality. Future studies which consider longer-term outcomes may need to consider the impact that stage differences may have on outcomes”

3. The authors conduct the analysis from a limited health system perspective – it would benefit from using the societal perspective, incorporating things like patients return to work, and QOL/survival beyond 12 weeks.

Although a societal perspective can be useful and informative for an economic evaluation a health system perspective is still of value. The cost-effectiveness evaluation in this study followed the National Institute of Health and Care Excellence (NICE) guidelines which suggest the reference base case for economic evaluation should be based on a health system perspective. In particular, NICE guidance states “productivity costs and costs borne by people using services and carers that are not reimbursed by the NHS or PSS should usually be excluded from analyses. That is, a societal perspective will not normally be used”.

Furthermore, NICE suggests that the base case for economic evaluation should take a time horizon 'long enough to reflect important differences in outcomes between the interventions being compared'. In this study we have identified very little difference in quality of life but larger differences in cost which are wholly driven by costs of the procedure's themselves rather than costs accrued in follow up. This suggests longer-term follow-up may be of limited value for the purposes of cost-effectiveness comparison from a health system perspective (the NICE reference case).

<https://www.nice.org.uk/process/pmg20/chapter/incorporating-economic-evaluation#the-reference-case>

4. It's a little surprising to see that the sensitivity analysis didn't include variation on PROMs or instrument costs (which have ranges in the provided data), especially since this is observational data. This would be a valuable addition, either as one-way or multi-way sens analyses:

Sanders GD, Neumann PJ, Basu A, Brock DW, Feeny D, Krahn M, Kuntz KM, Meltzer DO, Owens DK, Prosser LA, Salomon JA, Sculpher MJ, Trikalinos TA, Russell LB, Siegel JE, Ganiats TG. Recommendations for Conduct, Methodological Practices, and Reporting of Cost-effectiveness Analyses: Second Panel on Cost-Effectiveness in Health and Medicine. *JAMA*. 2016 Sep 13;316(10):1093-103. doi: 10.1001/jama.2016.12195. Erratum in: *JAMA*. 2016 Nov 8;316(18):1924. PMID: 27623463.

We agree that it is important to conduct sensitivity analysis in order to assess the impact of different measures and assumptions. As such we have performed and illustrated the results of one-way sensitivity analysis varying the number of procedures that are assumed to occur in a given year and the changes this has on the 'per surgery' cost of capital equipment and subsequent impact on the cost-effectiveness evaluation. We chose this one-way sensitivity analysis due to the very high cost of the robotic device. Variation of number of procedures would vary capital costs which were a component of the equipment costs found in table 2 and subsequently total procedure cost. Further to this, each analysis (including the base-case analysis) contains probabilistic sensitivity analysis through the inclusion of bootstrapping, with results reflected in the cost-effectiveness planes and confidence intervals around the cost and QALY differences. Bootstrapping was chosen as a technique which allows us to account for second order uncertainty around total costs and outcomes. We chose not to conduct one-way sensitivity analysis with QALYs or include changes in QALYs as part of a two-way sensitivity analysis due to the very small estimated QALY difference between the two surgical procedures. Given the small QALY differences identified between RALS and CLS it would, in our view, unlikely be of any benefit to perform one or two-way sensitivity analyses varying this outcome (e.g. it is unlikely that the direction or even magnitude of results would change significantly as estimated QALY differences are of such a small amount, with cost differences being the driver of variation between RALS and CLS in our cohort).

Minor Concerns:

1. In the introduction the authors reference the ASTEC study, stating that a 3rd of conventional laparoscopic surgery patients in that study had the procedures converted to open surgery. I believe the authors excluded converted to open operations, but it would be good to include these rates. It would be interesting if the authors could comment on if there is any evidence that these conversions could have been prevented with the use of robotics given their finding that the robotic cases in their analysis were patients with cancers at higher grade and stage.

We do not have available information around conversion to open surgery except to say that those included in our cohort did not have conversions recorded. However, we have added a section in to the limitations section to address this as a potential limitation of the cost-effectiveness analysis and a suggestion that any future studies in this area should fully account for conversions. We also note that published meta-analysis evidence suggests that RALS has a lower conversion rate than CLS and as such we can at least infer that accounting for conversions may increase the cost-effectiveness case for RALS (e.g., through higher resource us and subsequent costs from conversions).

In the limitation section we have added the following:

“This study did not include any cases which required conversion to open surgery. Future cost-effectiveness studies would benefit by accounting for these cases as they may impact cost-effectiveness results. A meta-analysis of robotic surgery suggests that RALS for endometrial cancer offers lower conversion rates than CLS and as such it may be inferred that accounting for conversion cases may further increase the cost-effectiveness of RALS versus CLS.”

2. In the introduction there is a reference that “robotic consumables are excluded from tariff”. Given the international readership of this journal, recommend expanding the explanation as readers from outside the UK may not understand what this means for robotics usage.

We have clarified this with the addition of a brief explanation as you have suggested.

“For now, NHS England have routinely commissioned RALS for the treatment of prostate cancer but for all other RALS services, robotic consumables are excluded from NHS tariff, which is a process in which NHS providers in England received healthcare funding “.

3. Remove “Functional Capacity” from the objective aims as the paper does not discuss this that much and could take away from the strength of the cost-effectiveness analysis.

Thank you for your advice, we have removed reference to functional capacity in the aims and objectives.

4. Table 1 and Supplement 4 are the same. Take our supplement 4

Supplemental table 4 has been removed as per your suggestion and supplemental material amended accordingly.

5. Add a description of what “Willingness to pay” means. This was unclear and confusing, taking away from the strength of the analysis

We have expanded this in the methods section (page 12) to state the following:

“For outcomes in which an intervention is both more costly and provided more QALYs can be further evaluated through consideration of willingness to pay, the additional cost willing to be paid in order to gain an additional QALY. To do this, a cost-effectiveness acceptability curve (CEAC) was produced which shows the number of bootstrapped iterations in which RALS was considered cost-effective over CLS across a range willingness to pay of threshold values (£0 to £50 000 per additional QALY gained).”

6. What is the National Institute for Health and care excellence threshold (again, for international readers)

We have amended a sentence in the cost-effectiveness methods section (page 12) to clarify the NICE threshold is usually between £20,000 to £30,000 per QALY

“In England, NICE typically use willingness to pay threshold values of £20 000 to £30 000 per QALY³⁵.”

Response to Reviewer: 2

1. Did you perform a power analysis? Do you have enough patients in the study to determine differences in HRQOL? What is the base population for this cohort (how many procedures in total during this time in the 12 NHS centres)? Please elaborate since this might improve the full picture and validity.

Information around the power analysis has been added to the manuscript:

“Sample size was calculated for the primary outcome of this study, the health-related quality of life measured by the EORTC QLQ C-30. Minimum clinically important difference of the global health score of the EORTC QLQ C-30 was defined as a standardised mean difference of 0.3 SDs. With 90% power (beta) and a 90% confidence interval (alpha) it was anticipated that the study would require a

total of 138 patients in each arm (with a maximum unequal allocation ratio of 2:1). Allowing for 25% attrition, this study aimed to recruit 173 patients in each cohort. “

Furthermore, we have addressed the low numbers of patients in the CLS cohort in the limitation section:

“Although this study recruited a sufficient number of RALS patients, it struggled to recruit CLS patients and as such results of the primary analysis are underpowered. Attrition was lower than anticipated, however CLS still fell short of requirements. It should be noted however, that the cost-effectiveness analysis collected ‘real world evidence’ of costs and utilities for patients undergoing CLS and RALS, which will be informative and of great value to health care funders. Furthermore, cost-effectiveness analysis has long been recognised as being underpowered as a secondary outcome in many studies. The techniques used in this study follow recommended guidelines and focus on estimating the cost and effect differences alongside the likelihood of an intervention being cost-effective, rather than hypothesis testing, with consideration given to uncertainty around point estimates and comparisons.”

2. Please clarify “effectiveness” in the abstract.

Effectiveness is perhaps inappropriate here and we have replaced this with quality of life to represent the quality-of-life outcomes measures in the primary analysis.

3. Can you please motivate the choice of questionnaires? An explanation might further improve your manuscript.

The EORTC score and 6MWT are both validated and accepted tools for reporting quality of life and functional capacity respectively. We have added the following sentences in the methods section to make it clearer to the reader that these scores have been validated and are appropriate to use in this setting.

“The QLQ C30 summary score is a validated measure for assessing quality of life for cancer patients and it presented as recommended by the EORTC Quality of Life Group”

“The 6MWT is a sub-maximal test of aerobic capacity which requires the individual to walk as far as possible during six minutes around a 30-meter course and the distance covered is recorded. It is a validated measure which has been recommended for use in cancer patients. Furthermore, the 6MWT has been applied in a clinical setting in a number of conditions as well as pre- and post-operatively”

4. HRQOL-please revise spelling of HRQOL, QOL etc throughout manuscript.

Quality of life and Health related quality of life are now written in full throughout the manuscript to aid clarity.

5. The dropouts, what happened to them? Results section page 12, line 12-14 “A total of 275 patients recruited to this study, 29 patients were excluded as they did not have procedures recorded”. Please specify/define what this drop-out means. In the abstract it says “298 patients recruited with 159 RALS, 73 CLS eligible for analysis” Explain the difference in figures, 298 vs 275.

The 298 in the abstract is an error that should state 275, this has been corrected.

Regarding the 29 patients who did not have procedures recorded, limited information was available to suggest whether this was missing data or patients not receiving intervention (e.g., through patient refusal or medical reasons). Furthermore, these patients did not have follow-up data recorded. This could be an indication of a variety of issues from real world data collection e.g. patient surgeries may have been delayed or cancelled resulting in them dropping out of the study between recruitment and

procedure taking place. We have had limited success in our attempts to get further information regarding these patients and why this data is missing.

6. Could a definitive RCT with embedded full economic evaluation be performed elsewhere in the world? Suggestions on how to further analyze and evaluate your results.

It may be possible for a definitive RCT to be conducted elsewhere in the world but we do not know enough about the current health systems to suggest whether this would be possible. Following this study, we considered an RCT in England, however as mentioned in the limitations section, RALS has expanded across the NHS and makes conducting such a study in England very difficult. If other health systems have similarly adopted the use of RALS, they may have the same difficulty to do this as in England.

7. Introduction Page 5 Line 50-53 Abstract missing comma `robot allows the surgeon to dissect remove and reconstruct tissues in a way that has not been possible`

We have added a comma to correct this, it now reads as:

“The combination of a wristed instrument design and 3-D high-definition optics of the robot, allows the surgeon to dissect remove and reconstruct tissues in a way that is not possible with non wristed instruments”

VERSION 2 – REVIEW

REVIEWER	Simianu, Vlad V Virginia Mason Medical Center
REVIEW RETURNED	30-Aug-2023

GENERAL COMMENTS	The authors have thoroughly addressed my and other reviewer comments. I have no other concerns and recommend acceptance for publication.
--

REVIEWER	Johannesson, Ulrika Karolinska Institute
REVIEW RETURNED	30-Aug-2023

GENERAL COMMENTS	This a very interesting and highly relevant study. With corrections made and questions answered it is now acceptable for publication.
---